# Perinatal outcome of meconium stained amniotic fluid among labouring mothers at teaching referral hospital in urban Ethiopia

**Lemi Belay Tolu** [1]*, **Malede Birara**[1], **Tesfalem Teshome**[1], **Garumma Tolu Feyissa**[2]

1 Saint Paul's Hospital Millennium Medical College, Addis Ababa, Ethiopia, 2 Department of Health, Behavior, and Society, Jimma University, Jimma, Ethiopia

* lemi.belay@gmail.com

## Abstract

### Objective

To determine the perinatal outcome of labouring mothers with meconium-stained amniotic fluid (MSAF) compared with clear amniotic fluid at teaching referral hospital in urban Ethiopia.

### Methods

A prospective cohort study was conducted among labouring mothers with meconium-stained amniotic fluid from July 1 to December 30, 2019. Data was collected with pretested structured questionnaires. A Chi-square test used to check statistical associations between variables. Those variables with a p-value of less than 0.05 were selected for cross-tabulation and binary logistic regression. P-value set at 0.05, and 95% CI was used to determine the significance of the association. Relative risk was used to determine the strength and direction of the association.

### Result

Among 438 participants, there where 75(52.1%) primigravida in a stained fluid group compared to 112 (38.5%) of the non-stained fluid group. Labour was induced in 25 (17.4%) of the stained fluid group compared to 25(8.6%) of a non-stained fluid group and has a statistically significant association with meconium staining. The stained fluid group was twice more likely to undergo operative delivery compared with a non-stained fluid group. There were more low Apgar scores at birth (36.8% versus 13.2%), birth asphyxias (9% versus 2.4%), neonatal sepsis (1% versus 5.6%), neonatal death (1% versus 9%), and increased admissions to neonatal intensive care unit (6.2% versus 21.5%) among the meconium-stained group as compared to the non-stained group. Meconium aspiration syndrome was seen in 9(6.3%) of the stained fluid group.

**Data Availability Statement:** All relevant data are within the manuscript and its Supporting Information files.

**Funding:** The author(s) received no specific funding for this work.

**Competing interests:** The authors have declared that no competing interests exist.

## Conclusion

Meconium-stained amniotic fluid is associated with increased frequency of operative delivery, birth asphyxia, neonatal sepsis, and neonatal intensive care unit admissions compared to clear amniotic fluid.

## Introduction

Meconium stained amniotic fluid(MSAF) is usually seen in 12 to 16% of deliveries [1]. Meconium passage is less common before 37 weeks of gestational age and increases steadily with gestational age [2]. It may represent the normal gastrointestinal maturation, or it may indicate an acute or chronic hypoxic event, thereby making it a potential warning sign of a fetal Compromise [3, 4]. Though its controversial to differentiate physiologic or pathologic meconium staining of amniotic fluid, there are few shreds of evidence that indicates its association with increased meconium aspiration syndrome, operative delivery, respiratory distress, neonatal sepsis, need for resuscitation, neonatal intensive care admission, and low Apgar score [5–8]. Besides, infants born through a meconium-stained amniotic fluid are more likely to develop respiratory distress and are at increased risk of perinatal death [1, 9]. Meconium aspiration syndrome (MAS) is characterized by the presence of respiratory distress with radiographic evidence of aspiration pneumonitis in the presence of meconium-stained amniotic fluid [4, 10]. MAS occurs in about 5% of deliveries with meconium-stained amniotic fluid [11], and death occurs in about 12% of infants with MAS [12].

The evidence of poor perinatal outcome associated with meconium-stained amniotic fluid mandates a well-designed study. Still, there is no well-designed comparative study in our country in general and no study at all in our hospital on the subject matter. The present study was, therefore, aimed at determining perinatal outcomes among laboring mothers with MSAF compared with clear amniotic fluid at teaching referral hospital in urban Ethiopia.

## Materials and methods

This was a hospital-based prospective cohort study. The study was conducted at Saint Paul's Hospital Millennium Medical College (SPHMMC), Addis Ababa, Ethiopia from July 1 to December 30, 2019. SPHMMC is a tertiary teaching referral hospital providing maternity service. The current prospective cohort study was conducted to determine the perinatal outcome of meconium-stained amniotic fluid (MSAF) compared with clear amniotic fluid among pregnant mothers in labour at Saint Paul's Hospital Millennium Medical College.

All consented pregnant women in labour who had completed more than 37 weeks of gestation, with viable singleton pregnancies with cephalic presentations and with no known fetal congenital anomalies were included. Twin pregnancy was excluded because of difficulty to determine chorionicity in labour and finding chorionicity and gestational age-matched twins. Gestational age was calculated from reliable last normal menstrual period or early ultrasound done before 24 weeks and those with an unknown date or without early ultrasound were excluded. Those with MSAF were exposed group referred to as "Stained fluid group", and those with clear amniotic fluid were non-exposed groups referred to as "non-stained fluid group" in our study.

Meconium stained amniotic fluid is the exposure variable of interest and classified into three:

1. **Grade one meconium-stained liquor:** small amount of meconium diluted in a plentiful amount of amniotic fluid. The fluid has only a slightly greenish or yellowish discoloration.

2. **Grade two meconium-stained liquor:** moderate meconium staining, when there is a fair amount of amniotic fluid, but it is stained with meconium. In this case, it will be 'khaki green' or brownish.

3. **Grade three meconium-stained liquor:** heavy staining, when there is reduced amniotic fluid and a large amount of meconium, making the staining quite thick, with 'pea soup' consistency.

Outcome variables were: 1st and 5th minute Apgar score, MAS, Birth asphyxia, NICU admission, early-onset neonatal sepsis (EONS), early neonatal death (END), and Operative delivery (CS or instrumental delivery). Covariates were parity, mode of delivery, duration of labour, duration of rupture of membrane, obstetric or medical complications like antepartum hemorrhage, pregnancy-induced hypertension, growth restriction, oligohydramnios, intra-amniotic infection(chorioamnionitis) and diabetes.

## Operational definitions

Perinatal outcome: in our study were used to describe composite of outcome variables.

Perinatal mortality: in our study means neonatal death within seven days of post-natal life per thousands of live births because all antenatal still-birth was excluded in the study.

Early neonatal death (END): neonatal death within seven days of post-natal life.

Operative delivery: in our study is meant for cesarean section or instrumental (vacuum or forceps delivery).

Birth asphyxia: in our study defined as 1st and 5th minute Apgar score of less than six and clinical diagnosis of perinatal asphyxia (PNA) at NICU.

MAS: in our study is a clinical diagnosis of respiratory distress in a neonate born through MSAF with a sign of meconium aspiration and was diagnosis put at NICU.

Open Epi version 3 was used to calculate the sample size for a matched cohort study. If 5% of patients with MSAF will develop MAS and 0.1% of patients with clear amniotic fluid will develop MAS and selection of two, clear amniotic fluids for one MSAF. By using the power of 80% and CI of 95%, the calculated sample size was 399 and adding 10% non-response rate total sample size was 438, so 146 stained fluid was collected consecutively and 292 non-stained who was in labour at the same time and who has close gestational age within one week were selected for comparison.

Trained midwives collected the data at the labour ward in pretested proforma. Data about the mode of delivery and outcome of birth were collected upon delivery in terms of apparent health, 1st and 5th minute Apgar score, early neonatal death (END), and referral to neonatal intensive care unit (NICU). Neonates who were referred to NICU were checked for admission diagnosis to NICU and their outcome on the seventh postnatal day from the neonatal chart and NICU logbook and those who are not referred to NICU and discharged home safely with mother were checked with a cell phone interview with mothers.

Data were entered and analyzed using SPSS version 23. Chi-square test was used to check statistical associations between meconium-stained amniotic fluid and outcome variables and covariates. Outcome variables with P value less than 0.05 were selected, and cross-tabulation was done to determine the strength and direction of the association between meconium staining of amniotic fluid and each outcome variable. All covariates with P value less than 0.05 (covariates associated with exposure variable) were selected for binary logistic regression to determine their association with each outcome variable. Statistical significance of the

association between exposure and outcome variables were determined by a 95% confidence interval and p-value set at 0.05. Adjusted relative risk (RR) was used to determine the strength and direction of the association between exposure and outcome variables.

## Ethical consideration

Ethical approval was obtained from Saint Paul's Hospital Millennium Medical College ethical review committee. Written informed consent was obtained from patient and confidentiality was maintained during data collection, analysis, and interpretation. All the datasets used and/ or analyzed during the current study are included in the manuscript and supplementary material.

## Results

According to the statistics office of the hospital, nearly 50,000 attended antenatal care, and around 9000 deliveries were attended in 2019, 35% of births were by cesarean section. A total of 438 pregnant women were included in this study, with a response rate of 99.3%. Three of post-partum women couldn't be traced on the seventh postpartum day for a phone interview and were lost to follow up. The meconium was described as grade I 51 (35%) patients, grade-II in 48 (33%) patients, and grade-III 45 (32%) patients (Fig 1).

Among 144 of the stained fluid group, 129 (89.6%) women were of 20-35-year age-group compared to 251(86.3%) of the non-stained fluid group. There is no statistically significant difference in the sociodemographic characteristics of participants (Table 1).

All participants had antenatal care (ANC) follow up except one patient in the non-stained fluid group. Fetal heart rate (FHR) monitoring was done with continuous cardiotocography (CTG) in 286 (98.3%) of the non-stained fluid group and 142(98.6%) of the stained fluid group. There where 75(52.1%) primigravida in a stained fluid group compared to112 (38.5%)

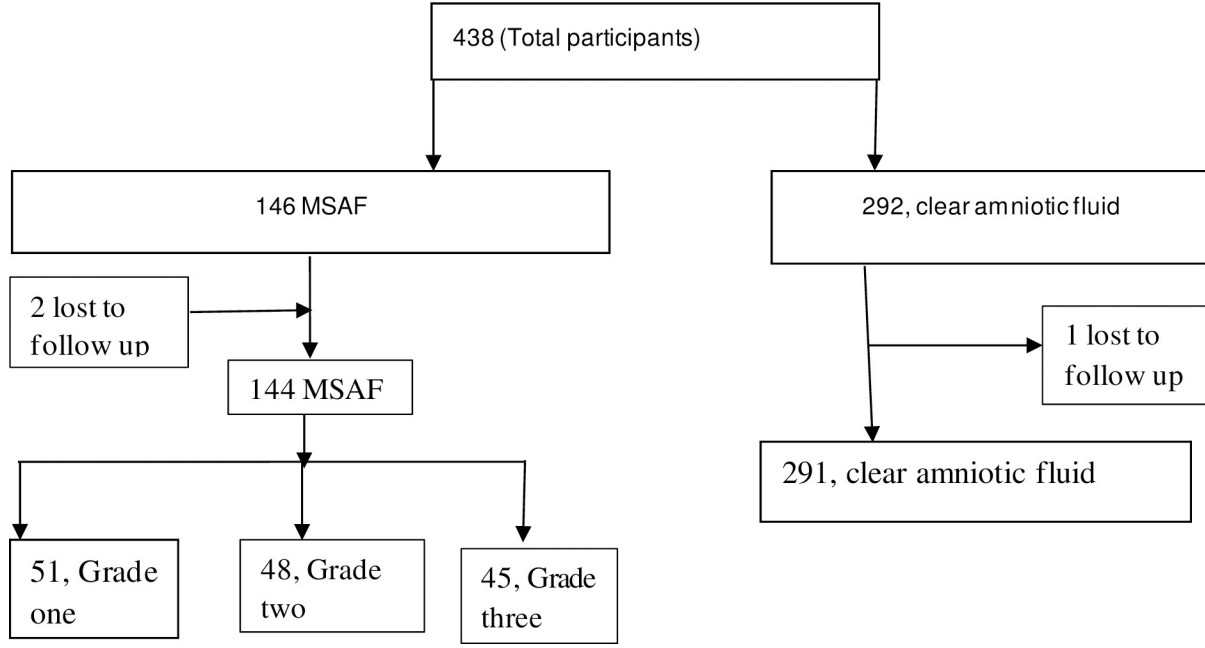

**Fig 1. Flow diagram of study participants.**

**Table 1. Sociodemographic characteristics of laboring mothers at SPHMMC from July 1 to December 30, 2018.**

| Variables. | Category. | Non-stained fluid. | | Stained fluid. | | Chi-square(p-value) |
|---|---|---|---|---|---|---|
| | | Frequency | % | Frequency | % | |
| Age | <20 | 18 | 6.2 | 10 | 6.9 | |
| | 20–35 | 251 | 86.3 | 129 | 89.6 | 2.802(0.241) |
| | >35 | 22 | 7.5 | 5 | 3.5 | |
| Level of education | Illiterate | 21 | 7.2 | 10 | 6.9 | |
| | primary school | 120 | 41.2 | 63 | 43.8 | 0.281(0.962) |
| | high school | 106 | 36.4 | 51 | 35.4 | |
| | College and above. | 44 | 15.1 | 20 | 13.9 | |
| Religion | Orthodox Christian | 173 | 59.5 | 96 | 66.7 | |
| | Protestant Christian | 45 | 15.5 | 10 | 6.9 | |
| | Muslim | 73 | 25.1 | 38 | 26.4 | 6.405(0.063) |
| Occupation status | Daily laborer | 3 | 1.0 | 0 | 0.0 | |
| | Student | 2 | 0.7 | 1 | 0.7 | |
| | Employed | 62 | 21.3 | 35 | 24.3 | 2.065(0.721) |
| | Housewife | 216 | 74.2 | 105 | 72.9 | |
| | Others | 8 | 2.7 | 3 | 2.1 | |
| Marital status | Single | 3 | 1.0 | 2 | 1.4 | |
| | Married | 287 | 98.6 | 140 | 97.2 | |
| | Divorced | 0 | 0.0 | 2 | 1.4 | 4.663(0.193) |
| | Widowed | 1 | 0.3 | 0 | 0.0 | |

of the non-stained fluid group and only 9 (6.3%) of them are gravida five and above compared with 13(4.5%) non-stained fluid group (Table 2).

Pregnancy-induced hypertension (PIH) was seen 26(49.1%) of the stained fluid group compared with 22(35.5%) of the non-stained fluid group. Labour started spontaneously in 266 (91.4%) of the non-stained fluid group compared with 119(82.6%) stained fluid group and induced in 25(17.4%) of the stained fluid group compared to 25(8.6%) of the non-stained fluid group. Induced labour is seven times more likely to have meconium-stained fluid compared to spontaneous onset of labour (Table 2). Prolonged rupture of membrane above 12 hours was seen in 32(19.3%) of the stained fluid group and 71(24.4%) of the non-stained fluid group. Duration of labour above 24 hours was seen in 11(7.6%) of the stained fluid group compared to13 (4.5%) of the non-stained fluid group. There was no statistically significant difference in terms of duration of the rupture of membrane and labour between the two groups (Table 2).

Cesarean section was the mode of delivery in 64(44.5%) stained fluid group as compared to 66(22.6%) non-stained fluid group. Sixteen (10.6%) of the stained fluid group had instrumental deliveries compared to 13(4.5%) non-stained fluid group. Merging cesarean and instrumental delivery as operative delivery; 80(55.6%) stained fluid group undergoes operative delivery compared to 79(27.1%) of the non-stained fluid group. All thin (grade one) stained fluid gave birth vaginally, while 80(86%) of thick (grade two and three) group underwent operative delivery. All cesarean section are emergency operations for the indication of thick meconium, fetal distress and poor progress of labour in 30,20 and 14 of stained fluid group respectively compared to poor progress of labour, previous scar in labour, fetal distress and active bleeding in 35,15,10 and 6 non-stained fluid respectively. The stained fluid group was twice more likely to undergo operative delivery compared with the non-stained fluid group (Table 3).

**Table 2. Antepartum and intrapartum events of laboring mothers at SPHMMC from July 1 to December 30, 2018.**

| Character | Non-stained fluid. | | Stained fluid. | | Chi square (P-value) |
|---|---|---|---|---|---|
| | Frequency | % | Frequency | % | |
| Attended Antenatal care (ANC) | 290 | 99.7 | 144 | 100 | 4.532(0.482) |
| Parity. | | | | | 14.682(0.005) |
| Primigravida | 112 | 38.5 | 75 | 52.1 | |
| Gravid two | 88 | 30.2 | 39 | 22.1 | |
| Gravid three | 48 | 16.5 | 18 | 12.5 | |
| Gravid four | 30 | 10.3 | 3 | 2.1 | |
| Gravid five and above. | 13 | 4.5 | 9 | 9.3 | |
| Duration of labour | | | | | |
| <12 hours. | 181 | 62.4 | 80 | 55.6 | |
| 12–24 hours. | 96 | 33.1 | 53 | 36.8 | 2.870(0.238) |
| >24 hours. | 13 | 4.5 | 11 | 7.6 | |
| Onset of labour | | | | | . |
| Spontaneous | 266 | 91.4 | 119 | 82.6 | |
| Induced | 25 | 8.6 | 25 | 17.4 | 7.283(0.007) |
| Duration of rupture of membrane. | | | | | |
| <12 hours. | 220 | 75.6 | 112 | 77.8 | |
| 12–24 hours. | 50 | 17.2 | 23 | 16.0 | 0.274(0.872) |
| >24 hours. | 21 | 7.2 | 9 | 6.3 | |
| An obstetric or medical complication | | | | | |
| Chorioamnionitis | 3 | 4.8 | 4 | 7.5 | |
| Abruptio placenta | 14 | 22.6 | 7 | 13.2 | |
| PIH* | 22 | 35.5 | 26 | 49.1 | 8.237(0.144) |
| Growth restriction | 11 | 17.7 | 7 | 13.2 | |
| Pregestational Diabetes Mellitus | 7 | 11.3 | 1 | 1.9 | |
| Oligohydramnios | 5 | 8.1 | 8 | 15.1 | |
| FHB** follow up method | | | | | |
| Continuous cardiotocography. | 286 | 98.3 | 142 | 98.6 | |
| Pinard fetoscope | 4 | 1.4 | 2 | 1.4 | 0.496(0.780) |
| Mixed both methods | 1 | 0.3 | 0 | 0.0 | |

*pregnancy-induced hypertension

** Fetal heartbeat.

Infants with MSAF had low 5th minute Apgar scores and 31(21.5%) stained fluid group needed intensive care unit admissions compared to 18(6.2%) of the non-stained group. Meconium aspiration syndrome was seen in 9(6.3%), stained fluid group. Neonates born to stained fluid were 2.5 times at risk of death in the first seven post-natal life as compared to those born to the non-stained fluid. Incidence of birth asphyxia, neonatal sepsis, and NICU admissions

**Table 3. Mode of delivery of laboring mothers at SPHMMC from July 1 to December 30, 2018.**

| Mode of delivery | Stained fluid. Frequency (%) | Non-stained fluid. Frequency (%) | Total. | RR (95% CI) |
|---|---|---|---|---|
| Vaginally delivery | 64(44.4) | 212(72.8) | 276 | 0.640(0.546–0.766) |
| Operative delivery. | 80(55.6) | 79(27.1) | 159 | 2.170(1.666–2.827) |
| Total. | 144(100) | 291(100) | 435(100) | |

**Table 4. Perinatal outcome of laboring mothers at SPHMMC from July 1 to December 30, 2018.**

| Parameter | Non-stained fluid | Stained fluid | Chi-square (P-value) | RR (95% CI) |
|---|---|---|---|---|
| | Frequency (%) | Frequency (%) | | |
| Meconium sucked from Oro-pharynx | 2(0.7) | 56(38.9) | 21.657(0.001) | 4.136(3.423–4.998) |
| 5th minute Apgar score (<7) | 40(13.7) | 53(36.8) | 30.475(0.001) | 2.142(1.669–2.748) |
| Perinatal asphyxia | 7(2.4) | 13(9) | 9.631(0.002) | 2.059(1.449–2.926) |
| Early-onset neonatal sepsis | 3(1) | 8(5.6) | 8.001(0.005) | 2.267(1.539–3.340) |
| Admission to NICU | 18(6.2) | 31(21.5) | 22.685(0.001) | 2.161(1.660–2.813) |
| Early neonatal death | 3(1) | 13(9) | 17.388(0.001) | 2.598(1.972–3.424) |

was statistically higher among babies born to the stained fluid as compared to those who were born to non-stained fluid (Table 4).

Meconium stained amniotic fluid has a positive clinically significant association with a primigravida, induction of labour (Table 2), and Operative delivery (Table 3). So binary logistic regression was done to see the effect of those independent variables which are associated with meconium staining on perinatal outcome. None of those independent variables has an association with perinatal outcome, and the adverse perinatal outcomes are solely associated with MSAF (Table 5).

## Discussion

In the current study, participants had similar baseline characteristics except for primigravida and induced labour which are associated more with MSAF. This might be because of slow and protracted progress of labour among primigravida's increasing the possibility of meconium development. Saunders et al. [13] reported that cesarean sections were performed twice as frequently in subjects with meconium-stained amniotic fluid. Naveen S et al. [14] also reported a cesarean section rate of 49.1% in MSAF. The current study also showed the stained fluid group

**Table 5. Binary logistic regression of variables associated with meconium staining and perinatal outcomes.**

| Perinatal outcome | Independent variable | P-value | RR (95% CI) |
|---|---|---|---|
| 5th minute Apgar score | Mode of delivery | 0.391 | 2.009(0.409–9.878) |
| | The onset of labour. | 0.815 | 0.865(0.090–52.439) |
| | Parity. | 0.773 | 1.023(0.124–3.336) |
| Early neonatal death | Mode of delivery | 0.998 | 2.583(0.007–73941) |
| | The onset of labour. | 0.496 | 3.407(0.100–11.590) |
| | Parity | 0.855 | 1.000(0,048–61.884) |
| Perinatal asphyxia | Mode of delivery | 0.998 | 7.443(0.704–4.951) |
| | The onset of labour. | 0.998 | 2.2730.412–368.583) |
| | Parity | 0.921 | 1.000(0.219–16.886) |
| Meconium aspiration syndrome | Mode of delivery | 0.998 | 1.000(0.257–2.917) |
| | The onset of labour. | 0.998 | 1.000(0.090–32.439) |
| | Parity | 0.921 | 1.000(0.072–63.987) |
| Early onset neonatal sepsis | Mode of delivery | 0.200 | 0.200(0.024–1.683) |
| | The onset of labour. | 0.953 | 0.953(0.140–6.483) |
| | Parity | 0.430 | 0.430(0.063–4.505) |
| NICU admission | Mode of delivery | 0.142 | 4.640(0.597–36.061) |
| | The onset of labour. | 0.225 | 2.509(0.568–11.079) |
| | Parity | 0.745 | 0.425(0.219–16.886) |

was two times more likely to undergo operative delivery compared to the non-stained fluid group. Our study also highlighted that 86% of the thick stained amniotic fluid group (grades two and three) undergoes operative delivery which indicates the risk of fetal heart rate abnormality with meconium staining and cesarean section being used as a rescue for infants who are about to develop MASThis is in line with other studies showing a higher risk of complications with thick meconium staining [5, 15].

In the current study, 36.8% stained fluid group had Apgar score less than seven, and stained fluid was 2.1 times more likely to have low 5th minute Apgar score compared to the non-stained fluid group. This is much lower than the study conducted by Sori DA et al. [16] at JUSH, which shows an Apgar score of less than seven in 88% of the exposed group. But the outcome is high compared to, Patil et al. [17] study, which reported that 19% of babies with MSAF had poor Apgar scores and other studies [9, 17]. The difference can be explained by the high incidence of operative deliveries for cephalon-pelvic disproportion (CPD) and non-reassuring fetal heart rate patterns in Sori DA et al. and the current study, which can be a cause and sign of intrauterine fetal distress and asphyxia, respectively.

Several investigators have demonstrated an association of meconium staining and poor perinatal outcome [1, 7, 18, 19]. There is no intrapartum death in our study, but the stained fluid group was at increased risk of END13 (9%) compared with non-stained fluid group 3(1%). PNA occurs almost twice in the stained fluid group (9%) compared to the non- stained fluid group (2.4%) and newborns born to MSAF were twice more likely to require NICU admission compared to those born to mothers with a clear amniotic fluid.

In the current research, MAS was diagnosed in 6.3% of babies in stained fluid group. MAS was diagnosed in 12.8% of babies born through MSAF in the study done by Patil et al. [17]. Meconium aspiration syndrome develops in only 2 of every 1000 live-born infants and 2% of those new-borns born through MSAF [20]. Ninety-five percent of infants with inhaled meconium clear the lungs spontaneously [9, 20]. A study by Sori DA et al. [16] showed meconium aspiration syndrome was diagnosed in 18.5% of the neonates born through MSAF, which is very high and they explained as the possibility of overdiagnosis since the diagnosis of MAS in their study was made only with clinical judgment without Chest X-ray. Moreover, the current study was conducted at tertiary hospital located in the capital city of Ethiopia compared to study by Sori DA et al. conducted at Jimma University Hospital which is serving the rural population. The possible limited access to health care in rural population might result in late presentation despite poor progress of labour resulting in high MAS. Therefore, since the current study was conducted at tertiary urban based hospital with relatively good obstetric care among relatively literate and wealth population with easy access to care might limit its generalizability to other centers and the wider population. This highlights that it is imperative to conduct multicenter study incorporating all levels of care.

## Conclusions

Meconium stained amniotic fluid is worrisome as it is associated with increased frequency of operative delivery, birth asphyxia, neonatal sepsis, and neonatal intensive care unit admissions compared to clear amniotic fluid which was seen in the current study. Therefore, management requires appropriate intrapartum care with a continuous or strict one to one fetal heartbeat follow up. Furthermore, knowing the high risk of early neonatal death we advise early postnatal follow up should be considered for infants born to mothers with thick MSAF. Finally, we recommend well-controlled studies comparing the perinatal outcome of thick and thin stained amniotic fluid to stratify management accordingly.

## Supporting information

**S1 Checklist. Strobe checklist: Describes a completed strobe checklist for an observational study.**
(DOCX)

## Acknowledgments

We thank Pre-Publication Support Service (PRESS) for providing pre-publication peer-review and copy editing of our manuscript.

## Author Contributions

**Conceptualization:** Lemi Belay Tolu.

**Data curation:** Lemi Belay Tolu.

**Formal analysis:** Lemi Belay Tolu.

**Investigation:** Lemi Belay Tolu.

**Methodology:** Lemi Belay Tolu.

**Project administration:** Lemi Belay Tolu, Malede Birara, Tesfalem Teshome, Garumma Tolu Feyissa.

**Resources:** Lemi Belay Tolu, Malede Birara, Tesfalem Teshome, Garumma Tolu Feyissa.

**Software:** Lemi Belay Tolu, Malede Birara, Tesfalem Teshome, Garumma Tolu Feyissa.

**Supervision:** Lemi Belay Tolu, Malede Birara, Tesfalem Teshome, Garumma Tolu Feyissa.

**Validation:** Lemi Belay Tolu, Malede Birara, Tesfalem Teshome, Garumma Tolu Feyissa.

**Visualization:** Lemi Belay Tolu, Malede Birara, Tesfalem Teshome, Garumma Tolu Feyissa.

**Writing – original draft:** Lemi Belay Tolu, Garumma Tolu Feyissa.

**Writing – review & editing:** Lemi Belay Tolu, Garumma Tolu Feyissa.

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
