## [Decision Letter · Decision Letter 0]

9 Jul 2020

PONE-D-20-13739

Perinatal Outcome of Meconium Stained Amniotic Fluid among labouring mothers at teaching referral hospital in urban Ethiopia.

PLOS ONE

Dear Dr. Tolu,

Thank you for submitting your manuscript to PLOS ONE. After careful consideration, we feel that it has merit but does not fully meet PLOS ONE’s publication criteria as it currently stands. Therefore, we invite you to submit a revised version of the manuscript that addresses the points raised during the review process.

We look forward to receiving your revised manuscript.

Kind regards,

Abhishek Makkar, M.D.

Academic Editor

PLOS ONE

Journal Requirements:

2.We noticed you have some minor occurrence of overlapping text with the following previous publication(s), which needs to be addressed:

https://www.jcdr.net/ReadXMLFile.aspx?id=3781

In your revision ensure you cite all your sources (including your own works), and quote or rephrase any duplicated text outside the methods section. Further consideration is dependent on these concerns being addressed.

Additional Editor Comments (if provided):

Dear Dr. Tolu,

Thanks for your submission. Please address both reviewer's comments and resubmit revised version by 8/21/2020. Looking forward to revision.

Reviewers' comments:

Reviewer's Responses to Questions

**Comments to the Author**

1. Is the manuscript technically sound, and do the data support the conclusions?

Reviewer #1: Yes

Reviewer #2: Partly

2. Has the statistical analysis been performed appropriately and rigorously? 

Reviewer #1: I Don't Know

Reviewer #2: I Don't Know

3. Have the authors made all data underlying the findings in their manuscript fully available?

Reviewer #1: Yes

Reviewer #2: Yes

4. Is the manuscript presented in an intelligible fashion and written in standard English?

Reviewer #1: Yes

Reviewer #2: No

5. Review Comments to the Author

Reviewer #1: Comments and Suggestions for Authors

A well thought study looking at the perinatal outcome of meconium stained amniotic fluid(MSAF) among laboring mothers at teaching hospital in Ethiopia. This study brings out important information regarding the outcomes of MSAF in Ethiopia.

Corrections:

Line 42 –Can be corrected as: Meconium stained amniotic fluid is usually seen in 12 to 16% of deliveries.

Line 47- Can be corrected as: Though its controversial to differentiate physiologic or pathologic meconium staining of amniotic fluid, there are few shreds of evidence that indicates its association (instead of associated) with increased meconium aspiration syndrome, respiratory distress, neonatal sepsis and perinatal mortality.

Line 77: Gestational age was (instead of is) calculated from reliable last normal menstrual period or early ultrasound done before 24 weeks and those with an unknown date or without early ultrasound were (where) excluded.

Line 168: All thin (grade one) stained fluid gave birth vaginally, while 80(86%) of thick (grade two and three) group underwent (instead of undergone) operative delivery.

Line 190 : Our study also highlighted that 86% of (thich )thick stained 190 amniotic fluid group (grade two and three) undergoes operative delivery which indicates the risk of fetal heart rate abnormality with meconium staining

Line 202 : Several investigators

Line 209 : In the current research, MAS was (has) diagnosed in 6.3 % a baby of the stained fluid group, which is 2.3 times compared to a non-stained fluid group

Table :2 P value is not in line for prim gravida and Induced labor.

Suggestions

1.It is unclear what the primary objective of this study was in the introduction. To determine the outcomes for physiologic v pathologic meconium staining or to compare MSAF outcomes with clear amniotic fluid. My recommendation is to have a clear objective in your introduction.

2.My suggestion is to compare Meconium aspiration syndrome between different types of MSAF and not with clear amniotic fluid and also to analyze the outcome differences between different types of MSAF

3. Were there any other conflicting variables like scheduled repeat C- section? Were those mothers excluded

4. My suggestion is also to compare the outcomes based on different Gestational age (Early term, term and post term).

Reviewer #2: Dear Editor,

Thank you for the opportunity to review this paper. The author present epidemiology of MAS in Ethiopia. There is paucity of any epidemiological data for Africa and this paper is worth publishing.

General notes: The text should be proofread for spelling, spacing and grammatical mistakes. The overall readability is borderline, but in several spots it is very difficult to understand what the authors mean. Please have the text proofread and edited. I appreciate the authors efforts to have the text proofread by a professional service, but it seems the editing service used by authors was substandard.

Specific comments:

Line 68 and 69 do not belong to methods since you report the results. Please move it to the result section.

Line 158. You define prolonged rupture of membranes as longer than 12h. It seems to be different from 18 hours used in the US. Can you explain somewhere the choice of this particular interval?

Lines 172-174. By definition, MSAF is one of the requirements to establish MAS diagnosis (your lines 105-106). How the patients with no MSAF were diagnosed with MAS?

In the description of demography (Table 1), the religion part might be misunderstood by some readers. Please specify if the Orthodox religion means Orthodox Judaism or Orthodox Christianity. I would spell out religions as Christian with subgroups of Eastern Orthodox and Protestant. Alternatively, I would spell out Orthodox Judaism and Christian (Protestant).

Table 2. Please specify if the diabetes was type one or pregnancy-induced.

It is interesting, that the follow-up was performed via a cell phone connection. Please, comment if the population of the woman included in the study or admitted to the hospital truly represent the population of Addis-Ababa. I would expect that only small proportion of the population could afford cell phones. Please clarify if I am wrong.

Good luck.

6. PLOS authors have the option to publish the peer review history of their article (what does this mean?). If published, this will include your full peer review and any attached files.

Reviewer #1: No

Reviewer #2: No

---

## [Author Response · Author response to Decision Letter 0]

9 Jul 2020

July 10, 2020

To: PLOS ONE Editor in chief.

Dear Editor in chief.

We would like to thank the editor and reviewers for their thoughtful review of the manuscript. They raise important issues and their inputs are very helpful for improving the manuscript. We agree with almost all their comments and we have revised our manuscript accordingly. We respond below in detail to each of the editor comments. We hope that you find our responses satisfactory and that the manuscript is now acceptable for publication 

Looking forward hearing from you soon

Sincerely,

Lemi B Tolu (MD, Assistant prof of obstetrics and gynecology).

Saint Paul’s Millennium Medical College (SPHMMC)

Department of Obstetrics and Gynecology

Addis Ababa, Ethiopia.

Email: lemi.belay@gmail.com

Dear editor 

Thanks for thoughtful review of the manuscript. Below is point by point response to raised concerns and how we changed the manuscript according to the comments.

#Editor 

Response: Dear editor thank you very much manuscript revised according to the journal requirement. 

Response: Thank you a lot, concern addressed (line 42-62)

Reviewer #1: 

A well thought study looking at the perinatal outcome of meconium stained amniotic fluid (MSAF) among laboring mothers at teaching hospital in Ethiopia. This study brings out important information regarding the outcomes of MSAF in Ethiopia.

Response: Thank you very much for energizing comment, below is the line by line response to concerns. 

#Corrections:

1. Line 42 –Can be corrected as: Meconium stained amniotic fluid is usually seen in 12 to 16% of deliveries.

Response: Corrected (Line 42)

2. Line 47- Can be corrected as: Though its controversial to differentiate physiologic or pathologic meconium staining of amniotic fluid, there are few shreds of evidence that indicates its association (instead of associated) with increased meconium aspiration syndrome, respiratory distress, neonatal sepsis and perinatal mortality.

Response: Corrected (line47)

3. Line 77: Gestational age was (instead of is) calculated from reliable last normal menstrual period or early ultrasound done before 24 weeks and those with an unknown date or without early ultrasound were (where) excluded.

Response: Corrected (line 85)

4. Line 168: All thin (grade one) stained fluid gave birth vaginally, while 80(86%) of thick (grade two and three) group underwent (instead of undergone) operative delivery.

Response: Corrected (Line 178)

5. Line 190: Our study also highlighted that 86% of (thich) thick stained 190 amniotic fluid group (grade two and three) undergoes operative delivery which indicates the risk of fetal heart rate abnormality with meconium staining

Response: Corrected (Line 199)

6. Line 202: Several investigators

Response: Corrected (Line 212) 

7. Line 209: In the current research, MAS was (has) diagnosed in 6.3 % a baby of the stained fluid group, which is 2.3 times compared to a non-stained fluid group

8. Table :2 P value is not in line for prim gravida and Induced labor.

Response: corrected (table 2)

#Suggestions

1.It is unclear what the primary objective of this study was in the introduction. To determine the outcomes for physiologic v pathologic meconium staining or to compare MSAF outcomes with clear amniotic fluid. My recommendation is to have a clear objective in your introduction.

Response: Thank you, suggestion addressed (68-69)

2.My suggestion is to compare Meconium aspiration syndrome between different types of MSAF and not with clear amniotic fluid and to analyze the outcome differences between different types of MSAF

Response: Dear reviewer thank you very much, the wrong comparison with clear amniotic fluid is corrected as Meconium aspiration syndrome was seen in 9(6.3%) stained fluid group (line 183-184). Unfortunately, among those 9 (6.3%) who developed MAS we didn’t collected the subgroups on thickness of amniotic fluid. 

3. Were there any other conflicting variables like scheduled repeat C- section? Were those mothers excluded

Response: Dear reviewer our population are those in labour, admitted to labour ward. 

4. My suggestion is also to compare the outcomes based on different Gestational age (Early term, term and post term).

Response: Dear reviewer we did gestational age matched data collection to control possibility of gestational age confounding the outcomes. 

Reviewer #2: 

#General notes: The text should be proofread for spelling, spacing and grammatical mistakes. The overall readability is borderline, but in several spots, it is very difficult to understand what the authors mean. Please have the text proofread and edited. I appreciate the authors efforts to have the text proofread by a professional service, but it seems the editing service used by authors was substandard.

Response: Dear reviewer, thank you very much. We reviewed the whole manuscript to address the spelling, spacing and grammatical mistakes concern. 

#Specific comments:

1. Line 68 and 69 do not belong to methods since you report the results. Please move it to the result section.

 Response: Concern addressed (line 76-77 and line 144-145) 

2. Line 158. You define prolonged rupture of membranes as longer than 12h. It seems to be different from 18 hours used in the US. Can you explain somewhere the choice of this interval?

Response: Dear reviewer yes, it is true different definitions is used for prolonged rupture of membrane. Pediatrics commonly use 18 hrs. in our hospital to consider for risky of neonatal sepsis. In Obstetrics 12 hrs. is commonly used 

3. Lines 172-174. MSAF is one of the requirements to establish MAS diagnosis (your lines 105-106). How the patients with no MSAF were diagnosed with MAS?

Response: Thank you the wrong comparison corrected as Meconium aspiration syndrome was seen in 9(6.3%) stained fluid group (line 183-184)

4. In the description of demography (Table 1), the religion part might be misunderstood by some readers. Please specify if the Orthodox religion means Orthodox Judaism or Orthodox Christianity. I would spell out religions as Christian with subgroups of Eastern Orthodox and Protestant. Alternatively, I would spell out Orthodox Judaism and Christian (Protestant).

Response: Thanks, concern addressed as Orthodox Christian (Table 1)

5. Table 2. Please specify if the diabetes was type one or pregnancy-induced.

Response: Thanks, concern addressed as pregestational DM (Table 2)

It is interesting, that the follow-up was performed via a cell phone connection. Please, comment if the population of the woman included in the study or admitted to the hospital truly represent the population of Addis-Ababa. I would expect that only small proportion of the population could afford cell phones. Please clarify if I am wrong.

Response: Dear reviewer currently most Ethiopians, even rural one had cell phone. The fact that our study is in Addis Ababa, capital city makes the follow up easy as our study population were from Addis Ababa or surrounding city.

---

## [Decision Letter · Decision Letter 1]

26 Aug 2020

PONE-D-20-13739R1

Perinatal Outcome of Meconium Stained Amniotic Fluid among labouring mothers at teaching referral hospital in urban Ethiopia.

PLOS ONE

Dear Dr. Tolu,

Thank you for submitting your manuscript to PLOS ONE. After careful consideration, we feel that it has merit but does not fully meet PLOS ONE’s publication criteria as it currently stands. Manuscript does need minor revision.

Please address all concerns raised by Reviewer 2.  We will look forward to your submission.

We look forward to receiving your revised manuscript.

Kind regards,

Abhishek Makkar, M.D.

Academic Editor

PLOS ONE

Reviewers' comments:

Reviewer's Responses to Questions

**Comments to the Author**

1. If the authors have adequately addressed your comments raised in a previous round of review and you feel that this manuscript is now acceptable for publication, you may indicate that here to bypass the “Comments to the Author” section, enter your conflict of interest statement in the “Confidential to Editor” section, and submit your "Accept" recommendation.

Reviewer #1: All comments have been addressed

Reviewer #2: All comments have been addressed

2. Is the manuscript technically sound, and do the data support the conclusions?

Reviewer #1: (No Response)

Reviewer #2: Partly

3. Has the statistical analysis been performed appropriately and rigorously? 

Reviewer #1: (No Response)

Reviewer #2: I Don't Know

4. Have the authors made all data underlying the findings in their manuscript fully available?

Reviewer #1: (No Response)

Reviewer #2: Yes

5. Is the manuscript presented in an intelligible fashion and written in standard English?

Reviewer #1: (No Response)

Reviewer #2: No

6. Review Comments to the Author

Reviewer #1: (No Response)

Reviewer #2: Dear Authors,

Sorry for delay with my review – it is crazy around here (in my part of the planet).

I would like to stress it again, I like the idea of your paper and this data should be published in the best possible form.

I am still not satisfied with the English part. I see the other reviewer helped you with some of that, but the text still needs a major proofreading. For example, your home city is misspelled on the first page. It is spelled as “Addis ababa” �. The title has words started with capital letters in random order. Chose one style. Capitalize them all or none. Obviously, the first word in the sentence and the name of the country should be capitalized, should you chose not capitalizing all the words. And so on and so on for the entire paper including papers.

Your result part of the summary is too descriptive. The readers will be looking for specific numbers of incidence of MAS in MSAF, Apgar scores, NICU admission rates and so on. Your description what was higher or lower is not enough. Please, provide the data of the incidence of MSAF and MAS in the summary. Please add the proportion of primigravida mothers in your population to the summary, if the space allows. Actually, you could cut out a big proportion of the statistics part form the summary to include more data.

It is interesting, that most of your patient population were primigravida. Perhaps you may explain this in discussion, why your group is so different from the rest of the country with average fertility rate over 4. Also, your group is very different form the rest of the country in regard of the literacy. You report that 6-8% of your mothers were illiterate, while in general illiteracy rate in Ethiopia is reported to be more than 50%. Perhaps you would like to mention this and explain why your group is different. Could it be explained by better education in Addis Ababa or just the population your hospital serves? My primary concern is that you group is not true representative of population of the country in terms of education, wealth and accessibility to health care. This in turn could result in different accessibility and desire to use the health care services. With all value of your data it is important to mention the limitations.

Perhaps I have missed it, but I was not able to see the reason for Cesarean section delivery in your patients. It is important to indicate whether it was scheduled or the obstetricians had to do it emergently for any reason. Perhaps you also could discuss if the Cesarean section delivery places a baby at risk of MAS or the Cesarean section serves as a rescue for infants who are about to develop MAS.

Table 4 – I do not see a reason for removal of the MAS incidence from this table. Isn’t it the main outcome?

Good luck.

7. PLOS authors have the option to publish the peer review history of their article (what does this mean?). If published, this will include your full peer review and any attached files.

Reviewer #1: No

Reviewer #2: No

---

## [Author Response · Author response to Decision Letter 1]

27 Aug 2020

August 27, 2020

To: PLOS ONE Editor in chief.

Dear Editor in chief.

We would like to thank the editor and reviewers for their thoughtful review of the manuscript. They raise important issues and their inputs are very helpful for improving the manuscript. We agree with almost all their comments and we have revised our manuscript accordingly. We respond below in detail to each of the editor comments. We hope that you find our responses satisfactory and that the manuscript is now acceptable for publication 

Looking forward hearing from you soon

Sincerely,

Lemi B Tolu (MD, Assistant prof of obstetrics and gynecology).

Saint Paul’s Millennium Medical College (SPHMMC)

Department of Obstetrics and Gynecology

Addis Ababa, Ethiopia.

Email: lemi.belay@gmail.com

Dear editor 

Thanks for thoughtful review of the manuscript. Below is point by point response to raised concerns by reviewer#2.

Reviewer #2: Dear Authors,

1. I would like to stress it again, I like the idea of your paper and this data should be published in the best possible form.

Response: dear reviewer thanks a lot. We will also do our best to extent of our potential and consult people in our network. 

2. I am still not satisfied with the English part. I see the other reviewer helped you with some of that, but the text still needs a major proofreading. For example, your home city is misspelled on the first page. It is spelled as “Addis ababa” �. The title has words started with capital letters in random order. Choose one style. Capitalize them all or none. Obviously, the first word in the sentence and the name of the country should be capitalized, should you chose not capitalizing all the words. And so on and so on for the entire paper including papers.

Response: Dear Reviewer thanks for the comment. Title edited accordingly and the whole text proofreading done by two authors (me and Garumma Tolu Feyissa). However, “Addis Ababa” is normally spelled like that though we might find “Addis Abeba” in some texts.

3. Your result part of the summary is too descriptive. The readers will be looking for specific numbers of incidence of MAS in MSAF, Apgar scores, NICU admission rates and so on. Your description what was higher or lower is not enough. Please, provide the data of the incidence of MSAF and MAS in the summary. Please add the proportion of primigravida mothers in your population to the summary, if the space allows. You could cut out a big proportion of the statistics part form the summary to include more data.

Response: Dear reviewer the abstract section was edited accordingly (lines 31-40)

4. It is interesting, that most of your patient population were primigravida. Perhaps you may explain this in discussion, why your group is so different from the rest of the country with average fertility rate over 4. Also, your group is very different form the rest of the country in regard of the literacy. You report that 6-8% of your mothers were illiterate, while in general illiteracy rate in Ethiopia is reported to be more than 50%. Perhaps you would like to mention this and explain why your group is different. Could it be explained by better education in Addis Ababa or just the population your hospital serves? My primary concern is that you group is not true representative of population of the country in terms of education, wealth and accessibility to health care. This in turn could result in different accessibility and desire to use the health care services. With all value of your data it is important to mention the limitations.

Response: Dear reviewer thank you very much for the input, the concerns where addressed (lines 192-193 and lines 225-232). 

5. Perhaps I have missed it, but I was not able to see the reason for Cesarean section delivery in your patients. It is important to indicate whether it was scheduled, or the obstetricians had to do it emergently for any reason. Perhaps you also could discuss if the Cesarean section delivery places a baby at risk of MAS or the Cesarean section serves as a rescue for infants who are about to develop MAS.

Response: Included with indications for both groups (lines 171-175 and lines 198-199)

6. Table 4 – I do not see a reason for removal of the MAS incidence from this table. Isn’t it the main outcome?

Response: Dear reviewer MAS was removed from table because of the contradiction of MAS among clear amniotic fluid group and expressed in sentences (lines 178-179) as “Meconium aspiration syndrome was seen in 9(6.3%), stained fluid

---

## [Editor Report · Decision Letter 2]

4 Sep 2020

PONE-D-20-13739R2

Perinatal outcome of meconium-stained amniotic fluid among labouring mothers at teaching referral hospital in urban Ethiopia.

PLOS ONE

Dear Dr. Tolu,

Thank you for submitting your manuscript to PLOS ONE. After careful consideration, we feel that it has merit but does not fully meet PLOS ONE’s publication criteria as it currently stands. Therefore, we invite you to submit a revised version of the manuscript that addresses the points raised during the review process.

Concerns raised by reviewers still need to be addressed to present manuscript that is clear and easy to read. Your submission still requires substantial editing for English grammar and usage

We would recommend that you have your manuscript copy-edited by either a native-English speaking colleague or a professional copy-editing service. While you may approach any qualified individual or any professional scientific editing service of your choice, PLOS has partnered with American Journal Experts (AJE) to provide discounted services to PLOS authors. AJE has extensive experience helping authors meet PLOS guidelines and can provide language editing, translation, manuscript formatting, and figure formatting to ensure your manuscript meets our submission guidelines. If there are still language issues in text that AJE has edited, AJE will re-edit the text for free. To take advantage of this special partnership, visit the AJE website and enter referral code PLOS15 on the registration page for a 15% discount off AJE services (http://www.aje.com/c/plos15). If you are already registered with AJE, please log in and enter PLOS15 at the bottom of your researcher dashboard under ‘Join a Group.’ Please note that PLOS ONE does not receive any compensation in relation to services completed by AJE and that having the manuscript copyedited by AJE or any other editing services does not guarantee selection for peer review"

We look forward to receiving your revised manuscript.

Kind regards,

Abhishek Makkar, M.D.

Academic Editor

PLOS ONE

---

## [Author Response · Author response to Decision Letter 2]

15 Sep 2020

September 2020, 2020

To: PLOS ONE Editor in chief.

Dear Editor in chief.

We would like to thank the editor and reviewers for their thoughtful review of the manuscript. They raise important issues and their inputs are very helpful for improving the manuscript. We agree that the manuscript requires editing for English grammar and usage. Hence, the manuscript was copy edited by two authors (one research fellow at Drexel university/USA) and Pre-Publication Support Service (PRESS). PRESS is Michigan University/USA based publication support organization providing manuscript preparation and copy editing. We hope that you find our revision satisfactory and that the manuscript is now acceptable for publication 

Looking forward hearing from you soon

Sincerely,

Lemi B Tolu (MD, Assistant prof of obstetrics and gynecology).

Saint Paul’s Millennium Medical College (SPHMMC)

Department of Obstetrics and Gynecology

Addis Ababa, Ethiopia.

Email: lemi.belay@gmail.com

---

## [Editor Report · Decision Letter 3]

25 Sep 2020

PONE-D-20-13739R3

Perinatal outcome of meconium-stained amniotic fluid among labouring mothers at teaching referral hospital in urban Ethiopia.

PLOS ONE

Dear Dr. Tolu,

Thank you for submitting your manuscript to PLOS ONE. After careful consideration, we feel that it has merit but does not fully meet PLOS ONE’s publication criteria as it currently stands. Therefore, we invite you to submit a revised version of the manuscript that addresses the points raised during the review process.

I want to compliment authors for making significant improvement to last version. There are some minor issues that need addressed. Please address the following:

In Results section: Line 156: Either add  women or patients after pregnant.

Line 156: Add Patients after postpartum

Line 157: add and after interview.

Discussion:

236:you have typo, I think you mean to say non-stained

238: take out similar, its redundant.

239: Please rephrase the line, its not clear. Do you mean to state " 6.3% of babies in stained fluid group"

Conclusion: 260: I would advise making line more generalized than giving specific timeline, the way its worded is a strong statement to make. I am assuming you are concluding it based on limited data you mentioned in line 234 and 235 in discussion.

 Please consider changing it to " Knowing the high risk of early neonatal death we advise early postnatal follow up should be considered for infants born to mothers with thick MSAF."

We look forward to receiving your revised manuscript.

Kind regards,

Abhishek Makkar, M.D.

Academic Editor

PLOS ONE

---

## [Author Response · Author response to Decision Letter 3]

25 Sep 2020

September 25, 2020

To: PLOS ONE Editor in chief.

Dear Editor in chief.

We would like to thank the editor for the thoughtful review of the manuscript. They raise important issues and their inputs are very helpful for improving the manuscript. We agree with all points raised and modified the manuscript accordingly. We hope that you find our revision satisfactory and that the manuscript is now acceptable for publication 

Looking forward hearing from you soon

Sincerely,

Lemi B Tolu (MD, Assistant prof of obstetrics and gynecology).

Saint Paul’s Millennium Medical College (SPHMMC)

Department of Obstetrics and Gynecology

Addis Ababa, Ethiopia.

Email: lemi.belay@gmail.com

Dear Editor 

Thank you very much. Please see the point by point response to raised concerns 

1. In Results section: Line 156: Either add women or patients after pregnant.

Response: Done line 136 

Line 156: Add Patients after postpartum

Response: Done line 137

2. Line 157: add and after interview.

Response: Done line 137 

3. Discussion:236: you have typo, I think you mean to say non-stained

Response: thank you very much, yes, it is to mean non-stained, corrected line 212

4. 238: take out similar, its redundant.

 Response: Done, line 214

5. 239: Please rephrase the line, it’s not clear. Do you mean to state " 6.3% of babies in stained fluid group"

 Response: Done, line 215 to 216 

6. Conclusion: 260: I would advise making line more generalized than giving specific timeline, the way its worded is a strong statement to make. I am assuming you are concluding it based on limited data you mentioned in line 234 and 235 in discussion. Please consider changing it to " Knowing the high risk of early neonatal death we advise early postnatal follow up should be considered for infants born to mothers with thick MSAF."

Response: Dear editor thank you very much for such beautiful thought and input, it means a lot. Corrected as “knowing the high risk of early neonatal death we advise early postnatal follow up should be considered for infants born to mothers with thick MSAF”, line 236-237

---

## [Editor Report · Decision Letter 4]

26 Oct 2020

Perinatal outcome of meconium-stained amniotic fluid among labouring mothers at teaching referral hospital in urban Ethiopia.

PONE-D-20-13739R4

Dear Dr. Tolu,

We’re pleased to inform you that your manuscript has been judged scientifically suitable for publication and will be formally accepted for publication once it meets all outstanding technical requirements.

Kind regards,

Abhishek Makkar, M.D.

Academic Editor

PLOS ONE
---

## [Editor Report · Acceptance letter]

5 Nov 2020

PONE-D-20-13739R4 

Perinatal outcome of meconium stained amniotic fluid among labouring mothers at teaching referral hospital in urban Ethiopia. 

Dear Dr. Tolu:

I'm pleased to inform you that your manuscript has been deemed suitable for publication in PLOS ONE. Congratulations! Your manuscript is now with our production department. 

Kind regards, 

on behalf of

Dr. Abhishek Makkar 

Academic Editor

PLOS ONE